# The Use of Biochemical and Biophysical Markers in Early Screening for Preeclampsia in Mongolia

**DOI:** 10.3390/medsci6030057

**Published:** 2018-07-20

**Authors:** Urjindelger Tserensambuu, Ariunbold Chuluun-Erdene, Munkhtsetseg Janlav, Erkhembaatar Tudevdorj

**Affiliations:** 1National Center for Maternal and Child Health, Khuvisgalchdiin Street, Bayangol District, Ulaanbaatar 16060, Mongolia; ts_urjee@yahoo.com; 2Department of Biochemistry and Laboratory Medicine, School of Biomedicine, Mongolian National University of Medical Sciences, S.Zorig Street, Sukhbaatar District, Ulaanbaatar 14210, Mongolia; ariunbold@mnums.edu.mn; 3Department of Obstetrics and Gynecology, School of Medicine, Mongolian National University of Medical Sciences, S.Zorig Street, Sukhbaatar District, Ulaanbaatar 14210, Mongolia; t_erkhembaatar@yahoo.com

**Keywords:** mean arterial pressure, preeclampsia, pregnancy-associated plasma protein-A, placental growth factor

## Abstract

Preeclampsia (PE) is a major cause of maternal and perinatal morbidity and mortality, particularly in developing countries. In Mongolia, preeclampsia and eclampsia have occurred among pregnancy complications at a rate of 25% in recent years. Recent studies in the literature have screened for preeclampsia by combining maternal factors with biomarkers. This study was conducted using prospective cohort research including 393 singleton pregnancies at 11–13^+6^ weeks. Maternal plasmas pregnancy-associated plasma protein-A (PAPP-A) and maternal serum placental growth factor (PlGF) were measured using Perkin Elmer time-resolved fluoroimmunoassay (DELFIA) kits, and the measurement of mean arterial pressure (MAP) was performed by automated devices and the uterine artery pulsatility index was measured by Doppler ultrasound. In the study population, there were 16.7% showing complicated preeclampsia. The receiver-operating characteristics (ROC) curve analysis showed a sensitivity of 71.21%, and a specificity of 75.54% when the mean arterial pressure cut-off was 89.5 mm; while a sensitivity of 33.36% and specificity of 77.68% were observed when the uterine artery mean pulsatility index (mPI) cut-off was 2.34; a sensitivity of 79.66% and specificity of 44.04% were observed when the PAPP-A cut-off was 529.1 mU/L; and a sensitivity of 74.58% and specificity of 46.6% were observed when the PlGF cut-off was 39.87 pg/mL. The detection rates following the combination of markers with the maternal history were as follows: 62.7% with mean arterial pressure, 69.5–82.9% with two markers 86.5% with three markers and 91.4% with four markers. In conclusion, the mean arterial pressure was highly sensitive and demonstrated its easy usage and cost-effectiveness as a predictive marker for the early screening of preeclampsia from other biomarkers.

## 1. Introduction

Responsible for complications in 2–8% of overall pregnancies, preeclampsia (PE) has been the main reason for maternal and perinatal morbidity and mortality, especially in developing countries [1,2]. Preeclampsia and eclampsia have been found to be responsible for 25% of all pregnancy complications in Mongolia in recent years according to the Mongolian Health Development Center [3]. The country’s maternal mortality rate due to preeclampsia and eclampsia was 17.7% between 2012 and 2015 [4].

From the practices of the past 20 years, it has become clear that collecting and combining data from patient histories with the results of biophysical and biochemical tests starting from a mother’s first hospital visit for antenatal care can predict the patient-specific risks for all potential pregnancy complications, including fetal abnormalities, miscarriage and stillbirth, preeclampsia, preterm delivery, gestational diabetes, fetal growth restriction and macrosomia. With the early predication of patient-specific risks for these pregnancy complications, the outcome of the pregnancy would be significantly improved, transforming regular prenatal care visits into more patient-specific and disease-specific visits for positive outcomes [5].

The effective prediction of preeclampsia can be achieved with an initial assessment at 11–13 weeks of gestation with the combination of maternal characteristics, mean arterial pressure (MAP), uterine artery pulsatility index (UtA-PI), maternal serum placental growth factor (PlGF), and pregnancy-associated plasma protein-A (PAPP-A) [6,7,8].

Improved screening performance can be achieved by combining patient history with a series of biophysical and biochemical markers which present differently even in the first trimester of pregnancy in cases that subsequently develop PE. In the presence of PE, compared with unaffected controls, at 11–13 weeks’ gestation the UtA-PI and MAP and maternal serum or plasma levels of soluble endoglin (sEng), inhibin-A, activin-A, pentarxin-3 (PTX3), P-selectin are elevated, whereas serum PAPP-A, PlGF and placental protein-13 (PP13) are reduced [9,10,11]. Biophysical and biochemical markers are believed to be involved in placentation or in the chain of events that lead from impaired placentation to the development of the clinical symptoms of PE.

The evidence increasingly suggests that the gestational age at the start of the disease is directly related to both the degree of impaired placentation and the incidence of adverse fetal and maternal short and long-term consequences of PE [6].

Preeclampsia diagnosis should be presumed in the presence of high blood pressure (BP) and significant proteinuria in the second half of pregnancy in a previously normotensive patient. There are several second-trimester studies that have reportedly used BP measurement as a screening method for subsequent development of PE and consequently reported contradictory results with false positive rate (FPR) differing between 7% and 52% and detection rates (DRs) between 8% and 93% [12,13,14].

In this study, we aimed to predict the PE by using biophysical and biochemical markers including MAP, mPI, PlGF and PAPP-A at the first trimester.

## 2. Materials and Methods

A total of 429 singleton pregnancies were included in the prospective screening study. All women attended in the study were covered by antenatal care services at the National Center for Maternal and Child Health (NCMCH) between March 2015 and June 2017.

The study was approved by the Ethical Committee of Mongolian National University of Medical Sciences on 13 February 2015 (protocol No 7/3/ 201507).

The criteria for the selection of singleton pregnancies for the screening of PE included crown–rump length (CRL) at 45–84 mm and with no signs of miscarriage. A total of 36 pregnancies with major fetal abnormalities and those ending in termination, miscarriage and no follow-up were excluded from the study.

A questionnaire with 50 questions asking for maternal age, parity (nulliparous/multiparous), pregnancy methods (spontaneous/use of assisted reproductive technology (ART), smoking status (yes/no), medical history (including chronic hypertension, diabetes mellitus, kidney diseases), obstetric history (previous pregnancy with or without PE), family history of PE (mother and sister) and inter pregnancy interval (years) was distributed to the patients. 

Time-resolved fluoroimmunoassay (DELFIA) analyzer (Wallac Oy, Turku, Finland, Cat # B055-201, A098-201) was used for the measurement of maternal serum PAPP-A and PlGF. Collected blood samples were centrifuged, and the serum was extracted for storage at −80 °C for subsequent biochemical analysis.

The measurement of the MAP was made with validated automated devices (HEM-7120, Оmron, Japan). After the women were seated and allowed to rest for 3–5 min, normal (22 to 32 cm) adult cuffs were fitted to their both arms. This was repeated two times with 1 min break in between. The MAP was calculated with the formula $\mathrm{MAP}=DP+\frac{\left(SP-DP\right)}{3}$, where DP represents diastolic blood pressure and SP, systolic blood pressure. We calculated the final MAP as the average of all four measurements.

Uterine artery Doppler studies including pulsatility index were measured through trans-abdominal and trans-vaginal ultrasound (Voluson E8 BT10, MFG 2010, Riverside, CA, USA) examinations. As stipulated in Fetal Medicine Foundation (FMF), when carrying out Doppler studies, a sagittal section of the uterus was obtained, and cervical canal and cervical internal os were identified. Subsequently, the transducer was gently tilted from side to side, and color flow mapping was used to identify each uterine artery along the side of the cervix and uterus at the level of the cervical internal os. Pulsed wave Doppler imaging was used with the sampling gate set at 2 mm to cover the whole vessel, and care was taken to ensure that the angle of insonation was less than 30°. When three to five similar consecutive waveforms had been obtained, PI was measured. The uterine artery mPI was calculated by adding the right and left pulsatility index together, divided by two.

All ultrasound and Doppler studies were carried out by a doctor who had received the appropriate certificate of competence in the 11–13^+6^ week scans and Doppler study from The Fetal Medicine Foundation [15].

Preeclampsia was defined when diastolic blood pressure was ≥90 mmHg, systolic blood pressure was ≥140 mmHg, and proteinuria was ≥300 mg in a 24 h period, after 20 weeks of gestation based on the World Health Organization (WHO) reference practiced by the local physician [16].

Data on pregnancy outcome were collected from the hospital maternity records or from patients.

### Statistical Analysis

Continuous data were represented as mean ± SD (standard deviation). Statistical significance was evaluated by *t*-test to compare the preeclampsia and unaffected groups, while a Mann–Whitney test was used for variables not normally distributed. Categorical variables were compared using the chi-square test. Spearman correlation test was used to determine the relationship between biophysical markers MAP, mPI of uterine artery and gestational age. Diagnosing each biomarker for PE was performed by receiver-operating characteristics (ROC) curves.

Logistic regression analysis was performed to assess the effect of the maternal and pregnancy characteristics on the development of preeclampsia. Patient-specific risk = odds/(1 + odds), odds = e^Y^. Y is based on a combination of bio-markers derived by stepwise multiple logistic regression analysis where Y = predicted probability (MAP or PlGF, PAPP-A and mPI combined parameters) + (if chronic hypertension = 3.51) + (if body mass index (BMI) ≥30 kg/m^2^ = 2.65) + (if previous preeclampsia = 4.15). Screening for preeclampsia were also assessed using ROC analysis. Differences were considered statistically significant when *p* < 0.05. The data analysis was performed using SPSS 21.0 (IBM Corporation, Armonk, NY, USA) and MedCalc 17.0.4 version (MedCalc Software, Ostend, Belgium.

## 3. Results

Out of the total 393 pregnancies attended in the study, 66 (16.8%) cases experienced preeclampsia, whereas 327 (83.2%) cases were unaffected by preeclampsia. The percentages of preterm PE and at-term PE were recorded as 18 (27.3%), 48 (72.7%), respectively.

Maternal characteristics of the screened population are summarized in Table 1.

We determined the mean values of biochemical markers PAPP-A, PlGF and biophysical markers MAP, mPI in the study population with relevance to their gestational ages (Table 2).

The mean level of PlGF was 38.6 ± 19.6 pg/mL in the preeclampsia group and 45.1 ± 24.0 pg/mL in the unaffected group, whereas the level of PAPP-A was 366.1 ± 195.3 mU/L in the preeclampsia group and 633.6 ± 496.9 mU/L in the unaffected group, showing a statistically significant difference of *p* < 0.01 for PlGF and *p* = 0.003 for PAPP-A, respectively.

In contrast to the unaffected group, the PE group demonstrated a higher mean maternal age, body mass index, parity, level of smoking, MAP, mPI and lower mean PAPP-A, PLGF, delivery age and baby’s birth weight (Table 3).

Biomarkers including MAP, mPI, PAPP-A, PlGF were analyzed with J (Youden’s index) and ROC curve tests for risk factors of PE. Consequently, a sensitivity of 71.21% and specificity of 75.54% (J = 0.467; area under the curve (AUC) 0.792; *p* < 0.001) were observed when the MAP cut-off was 89.5 (95% confidence interval (CI): 81.8–91.8) mmHg; with a sensitivity of 33.36% and specificity of 77.68% (J = 0.12; AUC 0.621; *p* < 0.001) when the mPI cut off was 2.34 (95% CI: 1.8–3.0); a sensitivity of 79.66% and specificity of 44.04% (AUC 0.320; *p* = 0.001) when the PAPP-A cut off was 529.1 (95% CI: 195.9–746.2) mU/L; and a sensitivity of 74.58% and specificity of 46.69% (AUC 0.615; *p* = 0.003) were observed when the PlGF cut-off was 39.87 (95% CI: 30.3–45.7) pg/mL (Figure 1).

Independent risk factors influencing the development of PE were evaluated by logistic regression analysis (Table 4). Pregnant women with a previous history of PE (relative risk (RR) 5.81, *p* < 0.001), chronic hypertension (RR 7.06, *p* < 0.001), a birth interval greater than 10 years (RR 2.08, *p* < 0.033) and obesity (RR 3.87, *p* < 0.001) had a higher risk of developing PE compared to the unaffected group.

Body mass index, chronic hypertension, and previous preeclampsia were evaluated by multiple regression analysis as risk factors of PE. The detection rate was calculated based on a combination of the history of these risk factors and biophysical and biochemical markers. The detection rates were increased with addition of the number of markers combined with the maternal history. The detection rates for a 5% FPR following the combination of markers with the maternal history were as follows: 62.7% with one marker (MAP), 69.5–82.9% with two markers (MAP + mPI, MAP + PAPP-A, MAP + PlGF), 86.5% with three markers (MAP + PAPP-A + PlGF) and 91.4% with four markers (MAP + PAPP-A + PlGF + mPI). The detection rates were also estimated by an FPR of 10% in Table 5.

## 4. Discussion

Among the other factors that lead to PE, such as extreme weather conditions and the size of the population, etc., the main factor contributing to this rate may be the fact that the diagnosis of PE includes gestational hypertension with no proteinuria in local practice and the absence of relevant guidelines for the prevention of PE in high-risk pregnancies in the country. Over the past 20 years, in the most developed and developing countries, patient-specific risks for a wide spectrum of pregnancy complications, including fetal abnormalities, miscarriage, stillbirth, preeclampsia, preterm delivery, gestational diabetes, fetal growth restriction and macrosomia, have been identified with the help of data such as patient history and biomarkers collected and combined from antenatal care visits initiated at 11–13 weeks [5]. The global rate for the prevalence of PE is 2–8%, whereas the results of our study show 16.8%.

However, according to the WHO guidelines, PE screening is to be determined based on demography and medical history, and women at an increased risk are advised to take 80 mg aspirin daily from early pregnancy until delivery [16]. An aspirin for evidence-based preeclampsia prevention (ASPRE) study revealed that recommending prophylactic aspirin at 150 mg per night from the first trimester prevented preterm PE by 50% [8].

In Mongolia, maternal and pregnancy characteristics, free-β hCG, PAPP-A, nuchal thickness (NT) and nasal bone (NB) have been used in screening for Down syndrome since 2012. Although preeclampsia is presently identified only by maternal history and blood pressure by systolic or diastolic reading alone, and by uterine artery Doppler testing in the second trimester and clinical symptoms in Mongolian practice, this study initially used PAPP-A, PlGF, MAP and uterine artery Doppler testing, particularly using the mean pulsatility index for screening of preeclampsia in the first trimester.

Although the cause of preeclampsia has not been fully explained yet, studies are increasingly proving that it is directly related to placentation insufficiency [17,18,19,20]. Low levels of PlGF and PAPP-A proteins in pregnancy in weeks 8–14 suggests risks for pregnancy complications, especially preeclampsia, being small for gestational age, intrauterine growth restrictions and stillbirth [20,21].

According to the studies by Yuval et al., cases of intrauterine growth restriction (IUGR) (RR = 3.12), pregnancy hypertension (RR = 6.09), and spontaneous miscarriage (RR = 8.76) were observed when the level of PAPP-A was at <0.25 MoM (Multiple of Median) [22].

The findings of this study have shown results similar to those of other major studies that documented the development of preeclampsia as being linked to a first-trimester increase in uterine artery PI and MAP and a decrease in serum PlGF and PAPP-A, by using a mini-combined test [23].

According to the study by Spenser et al., where 4390 pregnant women were involved, the sensitivity of preeclampsia increased from 50% to 62% (with FPR of 5%) when a combination of decreased PAPP-A in the first trimester and increased uterine artery PI at 22–24 weeks of gestation were present [24].

The studies conducted by Martin A. et al., Parra M. et al. and Gomez O. et al. showing an overall sensitivity of 25% for the prediction of preeclampsia has been reported by a total of 4993 patients at 11–14 weeks, improving the prediction of early onset severe disease to about 60%, at a 5% cut-off [25,26,27]. In our study, the sensitivity of mUt.A-PI was 33.3% and the specificity was 77.7%.

In the study conducted by Pilalis A et al., it was reported that the use of combined data of uterine artery PI and the maternal history of preeclampsia was better compared with the use of the uterine artery Doppler result alone [28] while our study showed significantly higher mPI (*p* < 0.019) in the preeclampsia group compared with the unaffected group. 

A meta-analysis performed on 3300 cases of PE, with more than 60,000 women involved, demonstrated the MAP to be more predictive of PE among low-risk women in the first or second trimester in comparison to the either systolic or diastolic readings alone [29]. On the other hand, maternal characteristics combined with biophysical markers such as first trimester MAP and mPI were identified as a strong tool for the prediction of PE in the first trimester by some other reviews. Moreover, the incorporation of maternal characteristics and first trimester MAP resulted in a higher detection rate; a similar result to that revealed by other publications [29]. According to the study by Poon and Kametas, the detection rate of preeclampsia was 43.3% if maternal history was used alone, 37.5% for MAP alone and 62.5% for combined use [30]. In our study, the detection rate of preeclampsia with the use of combining maternal history and MAP was 62.7% (AUC = 0.773, *p* < 0.001). Furthermore, in our study, the MAP was considerably higher in the preeclampsia group (94.0 ± 59.05; 84.55 ± 8.15; *p* < 0.001) compared with the unaffected group.

In both PE and unaffected pregnancies, MAP and uterine artery PI are closely associated, which is why this correlation must be considered combined with these two biophysical markers in calculating the patient-specific risk for PE in order to avoid the overestimation of effects from each of the markers. The estimated detection rate of PE requiring delivery before 34, 37 and 42 weeks gestation in screening by maternal factors and biophysical markers are 80, 55 and 35%, respectively, at an FPR of 5 and 90%, 72 and 57%, respectively, at a FPR of 10% [7]. The detection rate (DR) drawn from our study with the use of maternal history, MAP and mPI was 69.5%. Early or late onset of PE was not taken into account in calculating this DR. Obesity served as a direct factor for developing PE (BMI > 30), which in our case was 3.87 (RR, 95% CI, *p* < 0.001) and it was found that a previous history of PE increased the risk for PE 5.81-fold (RR, 95% CI).

Among the advantages of MAP measurement, including easy usage and cost effectiveness, which are very important in developing countries, the biggest advantage is its higher sensitivity and predictability compared to the mPI of uterine artery. Therefore, we need to extend the knowledge of primary health professionals in using MAP as a strong predictor of preeclampsia.

The limitations of this study include the relatively low number of samples (393) compared to other studies (in European countries). However, the findings from our study proved the feasibility of using these biomarkers for the early screening of preeclampsia.

## 5. Conclusions

The MAP was a highly sensitive marker for the early screening of preeclampsia compared to the mPI of uterine artery at the first trimester.

The combined use of biomarkers such as PAPP-A, PlGF, MAP, and the mean pulsatility index of uterine artery for the detection of preeclampsia during early pregnancy increases the detection rate by up to 91.4%.

## Figures and Tables

**Figure 1 medsci-06-00057-f001:**
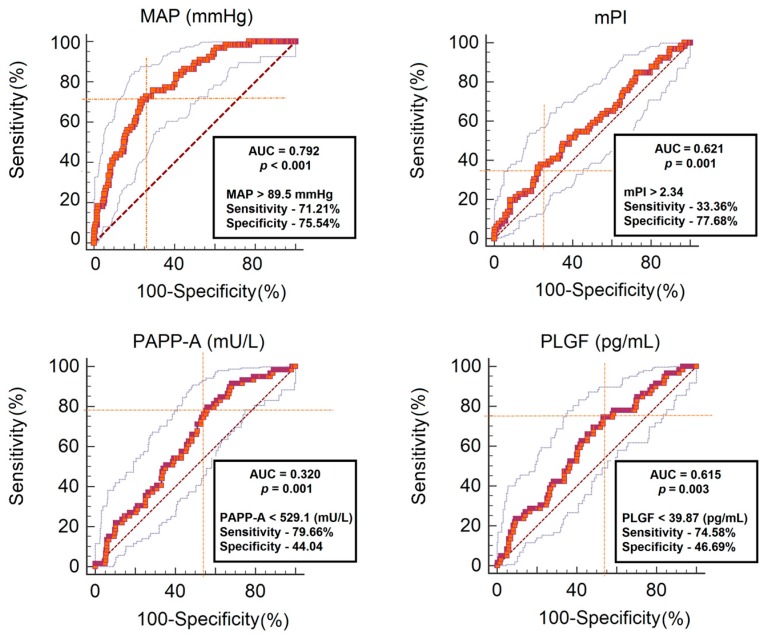
The best cut-off points of each biomarkers. PAPP-A—pregnancy-associated plasma protein-A; PlGF—maternal serum placental growth factor; MAP—mean arterial pressure; mPI—mean pulsatility index of uterine artery; AUC—Area Under a receiver-operating characteristics curve.

**Table 1 medsci-06-00057-t001:** Maternal characteristics in the screening population. Mean ± SD, *N* (%).

Characteristics	Total	Preeclampsia	Unaffected PE
Maternal age (years)	33.4 ± 6.0	36.1 ± 5.6	32.8 ± 5.9
Gestational age (weeks)	12.3 ± 0.6	-	-
Maternal weight (kg)	63.2 ± 10.8	69.2 ± 10.7	62.12 ± 10.6
Fetal crown-rump length (mm)	60.4 ± 8.6	-	-
BMI (kg/m^2^)	24.3 ± 3.9	26.8 ± 4.1	23.8 ± 3.7
<18.5	14 (3.6)	1 (1.5)	13 (3.9)
18.5–24.99	232 (59.0)	21 (31.8)	211 (64.5)
25.00–29.9	107 (27.2)	29 (43.9)	78 (23.8)
>30	40 (10.2)	15 (22.7)	25 (7.6)
Medical history			
Chronic hypertension	20 (5.1)	11 (16.7)	9 (2.8)
Renal disease	40 (10.2)	10 (15.2)	30 (9.2)
Diabetes II type	3 (0.8)	2 (3.0)	1 (0.3)
Normal	294 (74.8)	35 (53.0)	259 (79.2)
Other	34 (8.7)	8 (12.1)	28 (8.5)
Previous preeclampsia			
Yes	71 (25.9)	29 (58.0)	42 (19.1)
No	203 (74.1)	21 (42.0)	182 (80.9)
Family history of PE			
Yes	33 (8.4)	5 (7.6)	28 (8.5)
No	299 (76.1)	47 (71.2)	252 (77.1)
Unknown	61 (15.5)	14 (21.2)	47 (14.4)
Parity (*N* %)			
Nulliparous	119 (30.3)	16 (24.2)	103 (31.5)
Multiparous	274 (69.7)	50 (75.8)	224 (68.5)
Birth weight (grams)	3441.1 ± 503.5	3283.6 ± 633.2	3472.9 ± 467.8
GA at delivery (weeks)	39.0 ± 1.5	38.0 ± 1.8	39.2 ± 1.4

BMI—body mass index; PE—preeclampsia; GA—gestational age; SD—standard deviation.

**Table 2 medsci-06-00057-t002:** Biochemical and biophysical markers in gestational ages (Mean ± SD).

Weeks	PAPP-A (mU/L)	PlGF (pg/mL)	MAP (mmHg)	mPI
11–11^+6^	494.5 ± 443.4	37.8 ± 22.0	86.4 ± 9.9	2.1 ± 0.4
12–12^+6^	579.9 ± 497.1	45.8 ± 50.2	85.4 ± 8.2	2.0 ± 0.5
13–13^+6^	629.4 ± 519.8	63.2 ± 67.3	87.4 ± 9.6	1.9 ± 0.5

PAPP-A—pregnancy-associated plasma protein-A; PlGF—maternal serum placental growth factor; MAP—mean arterial pressure; mPI—mean pulsatility index of uterine artery.

**Table 3 medsci-06-00057-t003:** Maternal characteristics and biomarkers in study groups.

Characteristics and Markers	Preeclampsia	Unaffected PE	*p*-Value
Maternal age (year)	36.1 ± 5.6	32.8 ± 5.9	<0.001
BMI (Kg/m^2^)	26.7 ± 4.1	23.8 ± 3.7	<0.001
Parity (nulliparous/multiparous)	16/50	103/224	<0.001
Smoking (n) (yes/no)	11/55	39/288	<0.001
PAPP-A (mU/L)	366.1 ± 195.3	633.6 ± 496.9	0.003
PlGF (pg/mL)	38.6 ± 19.6	45.1 ± 24.0	0.01
MAP (mm Hg)	94.05 ± 9.05	84.55 ± 8.15	<0.001
mPI	2.16 ± 0.55	2.0 ± 0.50	0.019
Delivery age (weeks)	38.0 ± 1.8	39.0 ± 1.4	<0.001
Birth weight (grams)	3283.63 ± 633.19	3472 ± 467.86	0.005

**Table 4 medsci-06-00057-t004:** Independent risk factors of medical history for development of PE at the first trimester.

Risk Factors	Frequency *n* (%)		RR (95% CI)	*p*
	PE	Unaffected PE		
Previous PE	29 (58.0)	42 (19.1)	5.81 (3.27–11.16)	<0.001
Chronic hypertension	11 (16.7)	9 (2.9)	7.06 (2.79–17.84)	<0.001
Obesity	15 (22.7)	25 (7.3)	3.87 (1.89–7.92)	<0.001
Birth interval > 10 years	5 (7.8)	28 (8.6)	2.08 (1.03–4.19)	0.033
Kidney diseases (non glomerulonephritis)	10 (15.2)	30 (9.2)	1.76 (0.81–3.82)	0.143
Smoking	11 (16.9)	39 (11.9)	1.05 (0.72–3.12)	0.270

RR, Relative risk; 95% CI, confidential interval; PE, preeclampsia.

**Table 5 medsci-06-00057-t005:** Combination of biomarkers and detection rates.

	Detection Rate * (Sensitivity)	
	FPR (5%)	FPR (10%)
With History		
MAP	62.7 (40.0–80.8)	68.3 (46.3–86.5)
MAP + mPI	69.5 (48.7–83.1)	73.7 (54,7–89.6)
MAP + PAPP-A	79.3 (56.6–89.2)	81.1 (60.4–91.5)
MAP + PlGF	86.5 (73.4–90.2)	90.9 (78.5–92.3)
MAP + PlGF + PAPP-A + mPI	91.4 (78.0–96.0)	95.3 (80.1–96.5)

* Detection rate % (95% CI, confidence interval) for a fixed false positive rate (FPR); MAP: mean arterial pressure; mPI: mean pulsatility index; PAPP-A: pregnancy associated plasma protein A; PlGF: placental growth factor.

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
