# Peer review of "The Use of Biochemical and Biophysical Markers in Early Screening for Preeclampsia in Mongolia"

_medsci, 2018, doi:10.3390/medsci6030057_

Round 1
Reviewer 1 Report
This is an interesting paper about the detection rate of preeclampsia (PE) at 11-13 weeks, using maternal characteristics in combination with blood pressure and biochemical factors. Neverthereless, I have some questions:
The number is quite small, although the rate of PE is quite high, about 16%. Is it because this is a high risk population for PE? This should be addressed in the paper.
It should also be addressed if PE is at term, total, number of cases of preterm PE, < 34 weeks.
From my point of view, in the discussion part the authors should mention the necessity of screening for PE in the first trimester, to identify those patients at high risk of PE, in order to offer then a pharmacological intervention like aspirin 150 mg per day every night, from 12 weeks till 36 weeks. (ASPRE study).
References should be updated.
Author Response
Response to Reviewer 1 Comments
Point 1: The number is quite small, although the rate of PE is quite high, about 16%. Is it because this is a high risk population for PE? This should be addressed in the paper.
Response 1:
Current study was not intended to determine the prevalence of PE among Mongolian pregnant population, in this study were included pregnant subjects who undergone primary health examination and sent to advanced medical examination to the National Center for Maternal and Child Health (NCMCH). Therefore the number of sample size is not presenting the nation wide data and had higher rate of PE.
Point 2: It should also addressed if PE is at term, total, number of cases of preterm PE, <34 weeks.
Response 2:
We are analyzed data and inserted following sentences in the result section. The percentage of preterm PE and at term PE were recorded as 18 (27.3%), 48 (72.7%) respectively.
Point 3: From my point of view, in the discussion part the authors mention the necessity of screening for PE in the first trimester, to identify those patients at high risk of PE, in order to offer then a pharmacological intervention like aspirin 150 mg per day every night, from 12 weeks till 36 weeks (ASPRE study)
Response 3:
I agree with your point of view. I have tried to write about the point. We are aware of those guidelines and research findings including WHO, NICE and ASPRE for PE prevention to take low–dose aspirin (75 vs 150mg). However, the study we have done is intended to do PE screening in early stage of gestation to demonstrate the feasibility of using different markers in clinical settings. Some of those markers used for this study were applied in the country first time and showed their clinical significance for further use. The pharmacological interventions will be the next step of this study.
Reviewer 2 Report
The authors pointed that, country’s maternal mortality rate due to preeclampsia and eclampsia was 17.7% between 2012 and 2015 in Mongolia. What they want to say that preclampsia account for the 17.7% of the total mortality ratio, rate, percentage on totalmaternal mortality.. ? Althoough the methodology is correct, some points need to be adressed: Values of Sensitivity, Specificity should be given for a fixed FPR: 5 or 10% When given the AUC the 95%CI should be also given, to have a better idea of the performance for each marker.. It is clear that as much markers are in the equation the better the detection rate, as clearly shoown in table 5, but the confidence intervals of the AUC are not different. The test which formula will be the one that better suits for the population been more parsimonious and with a higher external validity, this is to use a model selection, such as Akaike, mallows Cp...Author Response
Response to Reviewer 2 Comments
Point 1: The authors pointed that, country’s maternal mortality rate due to preeclampsia and eclampsia was 17.7% between 2012 and 2015 in Mongolia. What they want to say that preeclampsia account for the 17.7% of the total mortality ratio, rate percentage on total maternal mortality?
Response 1:
17.7% is the rate percentage on total maternal mortality. It is not mortality ratio.
Point 2: Values of Sensitivity, Specificity should be given for fixed FPR: 5 or 10%
When given the AUC the 95% should be also given, to have better idea of the performance for each marker... It is clear that as much markers are in the equation the better the detection rate, as clearly shown in table 5, but confidence intervals of the AUC are not different. The test which formula will be the one that better suits for the population been more parsimonious and with a higher external validity, this is to use a model selection such as Akaike, mallows Cp…
Response 2:
1. According your recommendation, we added 95% CI to ROC analysis of each biomarkers in the interpretation of result.
2. About the 95% CI of AUC: these were little bit different each other due to little discrimination of all points in area of under curve of diagnosing separated biomarkers for preeclampsia (PE) except for mean arterial pressure (MAP) parameter. On the other hand, the abilities of the uterine artery pulsatility index (UtA-mPI), maternal serum placental growth factor (PlGF) and pregnancy-associated plasma protein-A (PAPP-A) parameter test to correctly classify those with PE and unaffected PE were less than MAP. But, the regression analysis of suitable model, were re-analysed according your recommendation above (model selection), has represented that predicted values of combined biomarkers indicates more sensitivity estimation to screen the PE significantly.
So, to avoid this drawback of 95% CI of AUC, we fixed table by ilustrating 95% CI of each sensitivity or detection rate in FPR 5 and 10%. preeclampsia
Reviewer 3 Report
In the present study, Tserensambuu and colleagues tested a combination of well established biochemical and biophysical markers of preeclampsia for the early diagnosis (11-13.6 weeks of gestation) of this severe syndrome in the Mongolian population. The Authors concluded that, among the biomarkers used, the Mean Arterial Pressure (MAP) was the most sensitive and cost effective for the early screening of Preeclampsia in the study population.
The study results are interesting, but there are some issues that the Authors must address before publication.
Major point
RESULTS
- Table 1. Patients with a medical history of Chronic Hypertension, Renal Disease or Diabetes Type II must be excluded from the study because their conditions clearly represent an important bias for preeclampsia diagnosis. At least, the Authors must consider and analyze these patients as separate groups and repeat data analysis;
Minor points
ABSTRACT
- Please explicit abbreviations in the abstract text (e.g. Mean Arterial Pressure - MAP);
INTRODUCTION
- Page 2 lines 44-47: The Author's state that "Effective prediction of preeclampsia can be achieved with initial assessment at 11-13 week’s of 45 gestation with the combination of maternal characteristics, mean arterial pressure (MAP), uterine 46 artery pulsatility index (UtA-PI), maternal serum placental growth factor (PlGF), and 47 pregnancy-associated plasma protein-A (PAPP-A)." Was this markers combination previously tested and scientifically validated for Preeclampsia diagnosis? If so, please report references.
- Please specify the study objective in the Introduction section;
MATERIALS AND METHODS
- Page 3, line 107: The software used is "SPSS" not "SPPP". Please correct;
ENGLISH
English must be revised.
Author Response
Response to Reviewer 3 Comments
Major point: Patients with medical history Chronic hypertension, Renal disease or Diabetes Type II must be excluded from the study because their conditions clearly represent an important bias for the preeclampsia diagnosis. At least, the authors must consider and analyze these patients as separate groups and repeat data analysis
Response 1:
We found several studies with similar study design, where subjects with mentioned conditions were included in those studies. As our study was implemented first time in our country, we used the study design of following studies.
1. Poon LC, Akolekar R, Lachmann R, Beta J and Nicolaides KH. Hypertensive disorders in pregnancy: screening by biophysical and biochemical markers at 11-13 weeks. Ultrasound Obstet Gynecol. 2010; 35: 662-670.Ã¥
2. Wright D, Syngelaki A, Akolekar R, Poon LC and Nicolaides KH. Competing risks model in screening for preeclampsia by maternal characteristics and medical history. Am J Obstet Gynecol. 2015; 213: 62 e1-10.
3. O'Gorman N, Wright D, Rolnik DL, Nicolaides KH, Poon LC. Study Protocol for the randomized controlled trial: combined multimarker screening and randomized patient treatment with ASpirin for evidence based PREeclampsia prevention (ASPRE). BMJ Open. 2016; 6: 6 e1-8.
4. Audibert F, Boucoiran I, An N, Aleksandrov N, Delvin E, Bujold E and Rey E. Screening for preeclampsia using first-trimester serum markers and uterine artery Doppler in nulliparous women. Am J Obstet Gynecol. 2010; 203: 383 e1-8.
5. W.Plasencia, N.Maiz, S.Bonino, C.Kaihura and K.H.Nicolaides. Uterine artery Doppler at 11+0 to 13+6 weeks in the prediction of pre-eclampsia. Ultrasound Obstet Gynecol 2007; 30:742-749
Minor points: Done
Round 2
Reviewer 3 Report
Reviewer's requests have been addressed.